# EVALUATION OF LARGE LANGUAGE MODELS VIA COUPLED TOKEN GENERATION

**Nina Corvelo Benz, Stratis Tsirtsis, Eleni Straitouri, Ivi Chatzi, Ander Artola Velasco, Suhas Thejaswi & Manuel Gomez-Rodriguez**
Max Planck Institute for Software Systems
Kaiserslautern, 67663, Germany
`{ninacobe,stsirtsis,estraitouri,ichatzi,avelasco,thejaswi,manuelgr}@mpi-sws.org`

## ABSTRACT

State of the art large language models rely on randomization to respond to a prompt. As an immediate consequence, a model may respond differently to the same prompt if asked multiple times. In this work, we argue that the evaluation and ranking of large language models should control for the randomization underpinning their functioning. Our starting point is the development of a causal model for coupled autoregressive generation, which allows different large language models to sample responses with the same source of randomness. Building upon our causal model, we first show that, on evaluations based on benchmark datasets, coupled autoregressive generation leads to the same conclusions as vanilla autoregressive generation but using provably fewer samples. However, we further show that, on evaluations based on pairwise comparisons, coupled and vanilla autoregressive generation can surprisingly lead to different rankings when comparing more than two models, even with an infinite amount of samples. To illustrate and complement our theoretical results, we conduct experiments with several large language models from the `Llama` family. We find that, in the MMLU benchmark dataset, coupled autoregressive generation requires up to $40\%$ fewer samples to reach the same conclusions as vanilla autoregressive generation. Further, in the LMSYS Chatbot Arena platform, we find that the win-rates derived from pairwise comparisons by a strong large language model to prompts differ under coupled and vanilla autoregressive generation.

## 1 INTRODUCTION

One of the most celebrated aspects of state of the art large language models (LLMs) is that they can solve open-ended, complex tasks across many different application domains such as coding, healthcare and scientific discovery Bubeck et al. (2023); Mozannar et al. (2024); Haupt & Marks (2023); Romera-Paredes et al. (2023). However, this is crucially what also makes the evaluation and comparison of LLMs very challenging—it is very difficult, if not impossible, to create a single benchmark. As a consequence, in recent years, there has been a flurry of papers introducing different benchmarks Bach et al. (2022); Wei et al. (2022); Talmor et al. (2019); Mishra et al. (2022); Chen et al. (2021); Liang et al. (2023); Longpre et al. (2023); Hendrycks et al. (2021b); Wang et al. (2023a); Ouyang et al. (2022); Wang et al. (2023b); Chiang et al. (2025); Taori et al. (2023); Zheng et al. (2023) In fact, one of the flagship conferences in machine learning has even created a separate datasets and benchmarks track!

In this context, it is somehow surprising that, in comparison, there has been a paucity of work understanding, measuring or controlling for the different sources of uncertainty present in the evaluations and comparisons of LLMs based on these benchmarks Miller (2024); Madaan et al. (2024); et al. (2024); Saad-Falcon et al. (2024); Boyeau et al. (2024); Chatzi et al. (2024b); Dorner et al. (2024); Gera et al. (2024). In our work, we focus on one source of uncertainty that has been particularly overlooked, the uncertainty in the outputs of the LLMs under comparison.

Given an input prompt, LLMs generate a sequence of tokens[1] as output using an autoregressive process Bengio et al. (2000); Radford et al. (2019). At each time step, they first use a neural network to map the prompt and the (partial) sequence of tokens generated so far to a token distribution. Then, they use a sampler to draw the next token at random from the token distribution.[2] Finally, they append the next token to the (partial) sequence of tokens, and continue until a special end-of-sequence token is sampled. To understand why, in the context of LLM evaluation and ranking, the above autoregressive process may lead to inconsistent conclusions, we will use a stylized example.

Consider we are given three LLMs $m_1$, $m_2$ and $m_3$, which we need to rank according to their ability to answer correctly two types of input prompts, $q$ and $q'$, picked uniformly at random. Assume that the true probability that each LLM answers correctly each type of input prompt is given by:

|  | $m_1$ | $m_2$ | $m_3$ |
|---|---|---|---|
| $q$ | 0.4 | 0.48 | 0.5 |
| $q'$ | 1 | 0.9 | 0.89 |

Then, one may argue that $m_1$ is the best LLM, followed closely by $m_3$, and $m_2$ is the worst, because the average probabilities that they answer a query picked uniformly at random correctly are 0.7, 0.695 and 0.69, respectively. However, if we conduct pairwise comparisons between outputs by two different LLMs to the same input prompt, as commonly done in practice, we may instead argue that $m_3$ is the best LLM, followed by $m_2$, and $m_1$ is the worst, because the probability that an LLM is preferred over others—the win-rates—are 0.16225, 0.15675, and 0.1545, respectively.[3] In our work, we argue that controlling for the randomization of the autoregressive processes underpinning the LLMs under comparison can, at least in certain cases, avoid such inconsistencies and lead to more intuitive conclusions. Along the way, we also show that it can reduce the number of samples required to reliably compare the performance of LLMs.

**Our contributions.** Our key idea is to couple the autoregressive processes underpinning a set of LLMs under comparison, particularly their samplers, by means of sharing the same source of randomness. To this end, we treat the sampler of each LLM as a causal mechanism that receives as input the distribution of the next token and the same set of noise values, which determine the sampler's (stochastic) state. By doing so, at each time step of the generation, we can expect that, if different LLMs map the prompt and the (partial) sequence of tokens generated so far to the same token distribution, they will sample the same next token. Loosely speaking, in the context of LLM evaluation and ranking, coupled autoregressive generation ensures that no LLM will have better luck than others. More formally, on evaluations based on benchmark datasets, we show that the difference in average performance of each pair of LLMs under comparison is asymptotically the same under coupled and vanilla autoregressive generation, but coupled autoregressive generation provably leads to a reduction in the required sample size. On evaluations based on (human) pairwise comparisons, we show that the win-rates of the LLMs under comparison can be asymptotically different under coupled and vanilla autoregressive generation and, perhaps surprisingly, the resulting rankings can actually differ. This suggests that the apparent advantage of an LLM over others in existing evaluation protocols may not be genuine but rather confounded by the randomness inherent to the generation process. To illustrate and complement our theoretical results, we conduct experiments with six LLMs of the `Llama` family. We find that, across multiple knowledge areas from the MMLU benchmark dataset, coupled autoregressive leads to a reduction of up to 40% in the required number of samples to reach the same conclusions as vanilla autoregressive generation. Further, using data from the LMSYS Chatbot Arena platform, we find that the win-rates derived from pairwise comparisons by a strong LLM differ under coupled and vanilla autoregressive generation. In Appendix 5, we include a comprehensive discussion of the limitations of our theoretical results and experiments, including additional avenues for future work. An open-source implementation of coupled autoregressive generation is available at `https://github.com/Networks-Learning/coupled-llm-evaluation`.

**Further related work.** Our work builds upon a very recent work on counterfactual token generation by Chatzi et al. (2024a), which also treats the sampler of an LLM as a causal mechanism. However,

---

[1]Tokens are the units that make up sentences and paragraphs, *e.g.*, (sub-)words, numbers, and special end-of-sequence tokens.

[2]If an LLM is forced to output tokens deterministically, multiple lines of evidence suggest that its performance worsens (Holtzman et al., 2020).

[3]Refer to Appendix C.6 for the detailed calculation of the average win-rates.

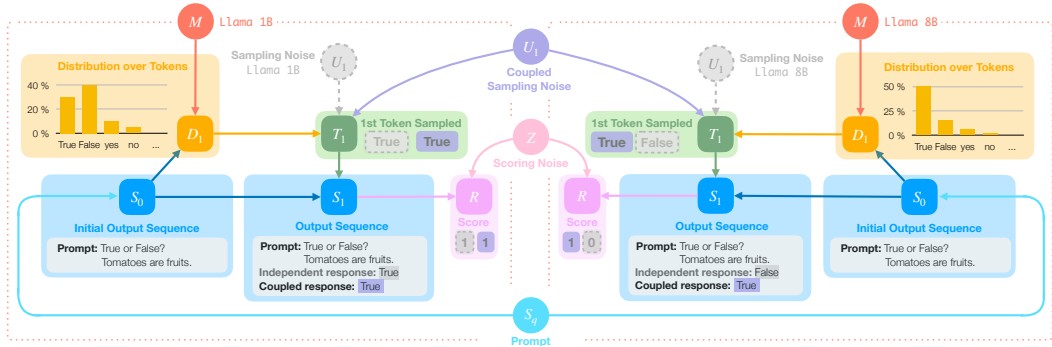

Figure 1: **Example of coupled autoregressive generation for `Llama 1B` and `Llama 8B`.** Boxes (circles) represent endogenous (exogenous) random variables. The value of each endogenous variable is given by a function of the values of its ancestors in the causal graph, as defined by Eq. 1. The value of the coupled noise variable $U_1$ (purple) is sampled independently from a given distribution $P_U$, and it determines the stochastic state of the samplers used by both `Llama 1B` and `Llama 8B` during the generation of token $T_1$.

their focus is different to ours; they augment a single LLM with the ability to reason counterfactually about alternatives to its own outputs if individual tokens had been different. Our work also shares technical elements with a recent work by Ravfogel et al. (2024), which develops a causal model to generate counterfactual strings resulting from interventions within (the network of) an LLM. However, their work does not study counterfactual generation for the purposes of model evaluation. In this context, it is also worth pointing out that the specific class of causal models used in the above works and ours, called the Gumbel-max structural causal model Oberst & Sontag (2019), has also been used to enable counterfactual reasoning in other domains (Tsirtsis et al., 2021; Noorbakhsh & Gomez-Rodriguez, 2022; Benz & Rodriguez, 2022).

Our work also builds upon the rapidly increasing literature on evaluation and comparison of LLMs Chang et al. (2024). Within this literature, LLMs are evaluated and compared using: (i) benchmark datasets with manually hand-crafted inputs and ground-truth outputs Bach et al. (2022); Wei et al. (2022); Talmor et al. (2019); Mishra et al. (2022); Chen et al. (2021); Liang et al. (2023); Longpre et al. (2023) and (ii) the level of alignment with human preferences, as elicited by means of pairwise comparisons Taori et al. (2023); Zheng et al. (2023); Li et al. (2024b;c); Boubdir et al. (2024); Singhal et al. (2023); Chiang et al. (2025). Within the literature on ranking LLMs from pairwise comparisons, most studies use the Elo rating system Askell et al. (2021); Dettmers et al. (2024); Bai et al. (2022); Wu et al. (2023); Lin & Chen (2023), originally introduced for chess tournaments Elo (1966). However, Elo-based rankings are sensitive to the order of pairwise comparisons, as newer comparisons have more weight than older ones, which leads to unstable rankings Boubdir et al. (2024). To address this limitation, several studies have instead used the Bradley-Terry model, which weighs pairwise comparisons equally regardless of their order Chiang et al. (2025); Boyeau et al. (2024). Nevertheless, both the Elo rating system and the Bradley-Terry model have faced criticism, as pairwise comparisons often fail to satisfy the fundamental axiom of transitivity, upon which both approaches rely Boubdir et al. (2024); Bertrand et al. (2023). Recently, several studies have used the win-rate Zheng et al. (2023); Chiang et al. (2025); Boyeau et al. (2024), which weighs comparisons equally regardless of their order and does not require the transitivity assumption. In our work, we focus on win-rates. However, we believe that it may be possible to extend our results to rankings based on Elo ratings and the Bradley-Terry model.

## 2 A CAUSAL MODEL FOR COUPLED AUTOREGRESSIVE GENERATION

Let $V$ denote a vocabulary (set) of tokens, including an end-of-sequence token $\perp$, $V^* = V \cup V^2 \cup \cdots \cup V^K$ be the set of sequences of tokens up to length $K$, and $\varnothing$ be the empty token.[4] An LLM $m \in \mathcal{M}$ takes as input a prompt sequence $s_q \in V^*$ and responds with an output sequence $s \in V^*$, generated using an autoregressive process. At each time step $i \in [K]$ of the process, the LLM first takes as input the concatenation of the prompt sequence $s_q$ and the (partial) output sequence $s_{i-1}$,

---

[4]Here, $V^j$ denotes the set of all sequences of length $j$ that can be constructed from the tokens in $V$. We restrict our attention to sequences of finite length ($\leq K$) because, in practice, the context window of LLMs is finite.

and generates a distribution over tokens $d_i \in \Delta(V)$. Then, it samples the next token $t_i \sim d_i$ from the distribution $d_i$ and creates the output sequence $s_i = s_{i-1} \circ t_i$, where $\circ$ denotes the concatenation of a token or sequence with another sequence. If $t_i = \bot$, it terminates and returns $s = s_i$, otherwise, it continues to the next step $i+1$ in the generation. Once the process is completed, the output sequence $s$ is assigned a score $r$.

Following the recent work by Chatzi et al. (2024a), we augment the above autoregressive process using a structural causal model (SCM) (Pearl, 2009; Peters et al., 2017), which we denote as $\mathcal{C}$. The SCM $\mathcal{C}$ is defined by the following structural equations:[5]

$$
S_0 = S_q, \quad D_i = \begin{cases} f_D(S_{i-1}, M) & \text{if } \texttt{last}(S_{i-1}) \neq \bot, \\ P_\varnothing & \text{otherwise} \end{cases}, T_i = \begin{cases} f_T(D_i, U_i) & \text{if } D_i \neq P_\varnothing, \\ \varnothing & \text{otherwise} \end{cases},
$$

$$
S_i = S_{i-1} \circ T_i, S = S_K, \quad \text{and } R = f_R(S, Z). \tag{1}
$$

In the above equations, $M, S_q, \boldsymbol{U} = (U_i)_{i \in \{1,...,K\}}$, and $Z$ are independent exogenous random variables, with $M \sim P_M$, $S_q \sim P_Q$, $U_i \sim P_U$, and $Z \sim P_Z$. Moreover, $f_D$, $f_T$ and $f_R$ are given functions, $P_\varnothing$ denotes the point mass distribution on $\varnothing$, and $\texttt{last}(S_{i-1})$ denotes the last token of the sequence $S_{i-1}$. Here, the function $f_D$ maps an input sequence $S_{i-1}$ to a distribution $D_i$ for the next token, using the architecture and network weights of the LLM $M$, the function $f_T$ and distribution $P_U$ specify the sampling mechanism that is used to sample the next token at each step of the generation process, following the distribution $D_i$, and the function $f_R$ and distribution $P_Z$ specify the exact scoring process by which the score $R$ is assigned to an output sequence $S$ during the evaluation of the LLM $M$.

Throughout the paper, we focus on sampling mechanisms that satisfy counterfactual stability (Oberst & Sontag, 2019; Tsirtsis et al., 2021; Chatzi et al., 2024a)—an intuitive form of consistency between the next token $T_i$, its distribution $D_i$, and the corresponding noise variable $U_i$.[6] Moreover, we allow the semantic meaning of the score $R$ as well as the support of its distribution to vary depending on the evaluation protocol. For example, in multiple-choice questions (Hendrycks et al., 2021a), $R \in \{0, 1\}$ may represent whether an LLM outputs a correct (1) or an incorrect (0) response. In pairwise comparisons (Chiang et al., 2025), $R \in \mathbb{R}^+$ may represent the level of user's satisfaction with the response provided by an LLM. Here, the variable $Z$ models any uncertainty in the scoring process, *e.g.*, in users' preferences (Bradley & Terry, 1952; Luce, 1959).

Building upon the above causal model, we can now formally express what it means to sample (and evaluate) output sequences by different LLMs using the same source of randomness,[7] a process we refer to as **coupled autoregressive generation**. Consider a specific model $m$, a prompt $s_q$, and fixed noise values $\mathbf{u}$ and $z$. It is easy to see that specifying these values is sufficient to (deterministically) specify and compute the exact value of the output sequence $S$ and its score $R$ using the autoregressive generation and scoring process given by Eq. 1. Then, we can formally express the coupled output sequences by two models $m$ and $m'$ and their corresponding scores as the result of *interventions* $do(M = m)$ and $do(M = m')$, respectively, where the $do(\cdot)$ operator forcibly sets the value of $M$ while keeping the prompt $s_q$ and the noise values $\mathbf{u}$, $z$ fixed (Pearl, 1994). In what follows, we denote the respective scores $R_m(\mathbf{u}, s_q, z)$ and $R_{m'}(\mathbf{u}, s_q, z)$, following standard notation (Pearl, 2009). For an illustration of coupled autoregressive generation against independent autoregressive generation—the vanilla approach—refer to Fig. 1.

In practice, one run of coupled autoregressive generation consists of two or more runs of autoregressive generation with the same prompt $s_q$ and noise values $\mathbf{u}$ and $z$, one per LLM.[8] From a causal perspective, we can view these runs as realizations of possible worlds where everything is equal except for the (architecture and network weights of the) LLM. Or we can also view one of these runs as a realization of the factual world and the other runs as realizations of different counterfactual worlds. Consequently, this lends support to attribute any difference in the scores $R_m(\mathbf{u}, s_q, z)$ across models

---

[5]Capital (lowercase) letters denote random variables (realizations).

[6]The default categorical sampler in `PyTorch` (Paszke et al., 2019), one of the most popular libraries used by state of the art LLMs, is an implementation of the Gumbel-Max SCM (Oberst & Sontag, 2019), which satisfies counterfactual stability. For a formal definition of counterfactual stability, refer to Appendix A.

[7]In our work, we implicitly assume that different LLMs share the same vocabulary $V$, however, in practice, this may not hold if the LLMs use different tokenizers. Refer to Appendix 5 for further discussion on this point.

[8]In practice, we may not always have control over the noise value $z$ (*e.g.*, when the scoring process is performed by an end user). However, even in such cases, we can still implement coupled autoregressive generation if the scoring processes occur simultaneously for each run, such as in pairwise comparisons.

$m \in \mathcal{M}$ to the models' architectures and weights rather than the randomness in their generation processes. Next, we will investigate the differences between coupled and independent autoregressive generation in evaluations based on benchmark datasets and pairwise comparisons.

## 3 THEORETICAL ANALYSIS

In this section, we focus on the evaluation and comparison of LLMs based on benchmark datasets, *e.g.*, multiple-choice questions (Hendrycks et al., 2021a), and theoretically investigate under which conditions coupled autoregressive generation requires fewer samples than independent autoregressive generation to reliably estimate the competitive advantage of one LLM over another. For further theoretical analysis of evaluations based on pairwise comparisons, refer to Appendix B.

Given a benchmark dataset characterized by an input prompt distribution $P_Q$, for each prompt $s_q \sim P_Q$, let $\mathrm{C}(s_q) \subset V^*$ denote the set of correct output sequences. Here, we consider binary scores $R_m(\mathbf{u}, s_q) = \mathbf{1}\{S_m(\mathbf{u}, s_q) \in \mathrm{C}(s_q)\} \in \{0, 1\}$, where $S_m(\mathbf{u}, s_q)$ denotes the output sequence of a model $m$ given a prompt $s_q$ under a realized sequence of noise values $\mathbf{u}$ and $\mathbf{1}\{\cdot\}$ is the indicator function.[9]

A standard approach to compare the performance of any pair of LLMs $m, m' \in \mathcal{M}$ using a benchmark dataset reduces to estimating the difference in their expected score, *i.e.*,

$$\mathbb{E}_{\boldsymbol{U}, \boldsymbol{U}' \sim P_{\boldsymbol{U}}, S_q \sim P_Q}[R_m(\underset{\uparrow}{\boldsymbol{U}}, S_q) - R_{m'}(\underset{\uparrow}{\boldsymbol{U}'}, S_q)], \tag{2}$$
$$\text{Independent generation}$$

where note that we use different noise variables $\boldsymbol{U}$ and $\boldsymbol{U}'$ for each LLM because, in the standard approach, each LLM generates outputs to each query independently (*i.e.*, using independent autoregressive generation). At first, one may think that coupled autoregressive generation will not be helpful. Under coupled autoregressive generation, the difference in the expected score adopts the following form:

$$\mathbb{E}_{\boldsymbol{U} \sim P_{\boldsymbol{U}}, S_q \sim P_Q}[R_m(\underset{\uparrow}{\boldsymbol{U}}, S_q) - R_{m'}(\underset{\uparrow}{\boldsymbol{U}}, S_q)] \tag{3}$$
$$\text{Coupled generation}$$

Therefore, based on the linearity of expectation and the fact that, under independent generation, both $\boldsymbol{U}$ and $\boldsymbol{U}'$ are sampled from the same distribution $P_{\boldsymbol{U}}$, it is easy to see that Eqs. 2 and 3 are equivalent. However, as we will show next, coupled autoregressive generation lets us reliably estimate the difference in the two LLMs' scores from finite samples faster. We first start by characterizing the relation between the variances of the difference of scores between LLMs using the following proposition (for the proofs of all propositions, refer to Appendix C):

**Proposition 3.1.** *For any pair of LLMs $m, m' \in \mathcal{M}$, it holds that*

$$\mathrm{Var}[R_m(\boldsymbol{U}, S_q) - R_{m'}(\boldsymbol{U}', S_q)] = \mathrm{Var}[R_m(\boldsymbol{U}, S_q) - R_{m'}(\boldsymbol{U}, S_q)] \\ + 2 \cdot \mathrm{Cov}[R_m(\boldsymbol{U}, S_q), R_{m'}(\boldsymbol{U}, S_q)] \tag{4}$$

This result immediately implies that, if the scores achieved by the LLMs under comparison are positively correlated, *i.e.*, the LLMs tend to generate a (in-)correct output sequence on the same prompts under the same noise values, then the variance of the difference in scores is lower under coupled generation than under independent generation, and thus we can expect a reduction in the sample size required to obtain equivalent estimation errors. In what follows, we analyze two canonical settings in which this condition holds and, in Section 4, we provide empirical evidence that, in a well-known benchmark dataset, this condition also holds.

In the first canonical setting, the correct response to each prompt is one of two given single-token sequences, the LLMs $m$ and $m'$ under comparison always output a response that is either of these two sequences, and the sampling mechanism used by the LLMs satisfies counterfactual stability. While this setting may seem restrictive, it is found in real-world scenarios. For example, think of true/false questions (or multiple-choice questions with two options) and evaluation protocols in which

---

[9]Our results can be extended to real-valued scores in a bounded interval.

the LLMs are explicitly instructed to always output true/false (or one of the two options) via their system prompt.[10] The following proposition shows that the variance of the difference in scores is lower under coupled autoregressive generation:

**Proposition 3.2.** *Consider a benchmark dataset such that* $\mathrm{C}(s_q) \subsetneq \{t_1, t_2\}$ *for all* $s_q \sim P_Q$, *where* $t_1$ *and* $t_2$ *are two single-token sequences. Let* $m$ *and* $m'$ *be two LLMs that assign positive probability to the sequences* $t_1$ *and* $t_2$ *and zero probability to any other sequence. If the sampling mechanism defined by* $f_T$ *and* $P_U$ *satisfies counterfactual stability, then, it holds that*

$$\mathrm{Var}[R_m(\boldsymbol{U}, S_q) - R_{m'}(\boldsymbol{U}', S_q)] > \mathrm{Var}[R_m(\boldsymbol{U}, S_q) - R_{m'}(\boldsymbol{U}, S_q)]. \tag{5}$$

In the second canonical setting, the correct response to each prompt is a single-token sequence, the LLMs $m$ and $m'$ under comparison always output a single-token response, and the sampling mechanism used by the LLMs is given by the Gumbel-Max SCM, in which

$$f_T(D_i, U_i) = \underset{t \in V}{\mathrm{argmax}} \left\{ \log\left(D_{i,t}\right) + U_{i,t} \right\},$$

where $U_{i,t} \sim \mathrm{Gumbel}(0,1)$ are i.i.d. noise variables associated with each token (Chatzi et al., 2024a). Similarly as in the first canonical setting, this second setting is also found in real-world scenarios—the default categorical sampler in the library `PyTorch` (Paszke et al., 2019) implements the Gumbel-Max SCM. The following proposition shows that, as long as $m'$ is *similar enough* to $m$, the variance of the difference in scores is lower under coupled generation:

**Proposition 3.3.** *Consider a benchmark dataset such that* $|\mathrm{C}(s_q)| = 1$ *for all* $s_q \sim P_Q$. *Let* $m$ *be an LLM that assigns positive probability to every single-token sequence and zero probability to any other sequence. If the sampling mechanism defined by* $f_T$ *and* $P_U$ *is given by the Gumbel-Max SCM, then, there exists a constant* $\varepsilon(m) > 0$ *such that, for every LLM* $m'$ *that assigns positive probability to every single-token sequence and zero probability to any other sequence and satisfies* $d(m, m') = \sup_{s_q} \|f_D(s_q, m) - f_D(s_q, m')\|_\infty < \varepsilon(m)$, *it holds that*

$$\mathrm{Var}[R_m(\boldsymbol{U}, S_q) - R_{m'}(\boldsymbol{U}', S_q)] > \mathrm{Var}[R_m(\boldsymbol{U}, S_q) - R_{m'}(\boldsymbol{U}, S_q)].$$

Based on the above proposition, we hypothesize that coupled autoregressive generation will reduce the number of samples required to reliably compare the performance of LLMs whenever these are sufficiently *similar*, *e.g.*, whenever we compare fine-tuned or quantized versions of the same pre-trained LLM.

## 4 EXPERIMENTS

In this section, we evaluate several large language models from the `Llama` family under coupled and independent autoregressive generation using: (i) the benchmark dataset MMLU Hendrycks et al. (2021a) and (ii) pairwise comparisons between outputs of the LLMs when prompted using open-ended questions from the LMSYS Chatbot Arena platform LMSYS (2023). In all our experiments, the LLMs use an implementation of the Gumbel-Max SCM Chatzi et al. (2024a) as a sampler both under coupled and independent autoregressive generation. For details on hardware, datasets and models used for experiments, refer to Appendix D.

### 4.1 EVALUATION ON THE MMLU DATASET

In this section, we compare three LLMs of different sizes, namely, `Llama-3.1-8B-Instruct` and `Llama-3.2-{1B, 3B}-Instruct`, using the MMLU benchmark dataset Hendrycks et al. (2021a), which comprise 14,042 multiple choice questions covering 52 knowledge areas. Recall that our theoretical results in Section 3 suggest that coupled autoregressive generation requires fewer samples than independent generation to reliably estimate the competitive advantage of one LLM against another in certain canonical settings. Here, our goal is to empirically investigate to what extent these results generalize to evaluations based on the MMLU dataset.

**Experimental setup.** In our experiments, for each multiple choice question in the MMLU benchmark dataset, we provide the question itself together with the available options (4 for each question,

---

[10]Here, our goal is to illustrate that there exist natural conditions under which coupled autoregressive generation is provably beneficial in comparison to independent autoregressive generation. However, in practice, in this canonical setting, one could directly use the LLMs' probabilities for the two tokens in each prompt to estimate the average difference of scores exactly.

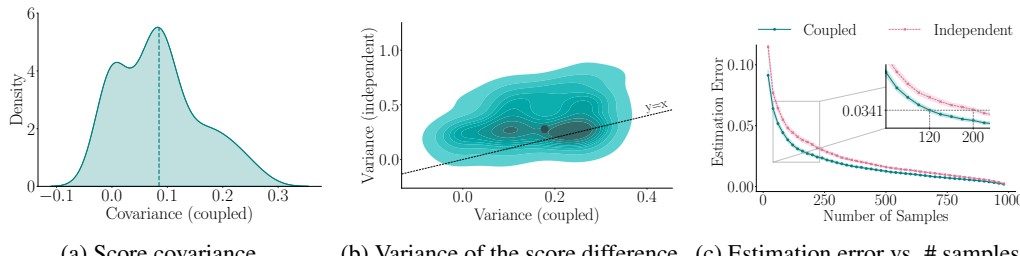

(a) Score covariance    (b) Variance of the score difference    (c) Estimation error vs. # samples

Figure 2: **Comparison between `1B` and `3B` on multiple-choice questions from the MMLU dataset.**
Panel (a) shows the kernel density estimate (KDE) of the covariance between the scores of the two
LLMs on each question under coupled generation; the dashed line corresponds to the average value.
Panel (b) shows the KDE of the variance of the difference between the scores of the LLMs on each
question under coupled and independent generation; the highlighted point corresponds to the median
value. Panel (c) shows the absolute error in the estimation of the expected difference between the
scores of the LLMs against the number of samples; for each point on the x-axis, we perform $1,000$
sub-samplings and shaded areas correspond to $95\%$ confidence intervals. Across all panels, we use
all questions from the knowledge area "college computer science" of MMLU. Refer to Appendix E
for qualitatively similar results for other LLMs.

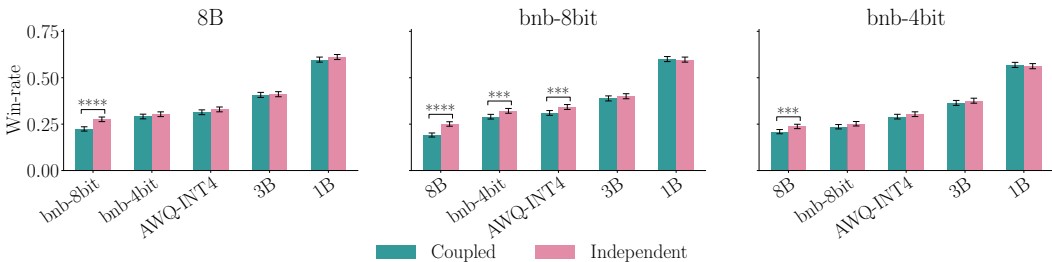

Figure 3: **Empirical win-rate of `8B`, `bnb-8bit` and `bnb-4bit` against any other LLM on
questions from the LMSYS-Chat-1M dataset.** Each empirical win-rate is computed using pairwise
comparisons between the outputs to $500$ questions with $10$ (different) random seeds under both
coupled and independent generation. The error bars correspond to $95\%$ confidence intervals. For each
pair of empirical win-rates under coupled and independent generation, we conduct a two-tailed z-test,
to test the null hypothesis that the empirical win-rates are the same; ($****$, $***$) indicate $p$-values
($< 0.0001$, $< 0.001$). Refer to Appendix F for qualitatively similar results for other LLMs.

indexed from A to D) as an input prompt to the LLMs. Further, we instruct the LLMs to generate an
output sequence comprising only the index of the selected option through a system prompt—refer
to Appendix D for the exact prompt. To evaluate the outputs provided by each LLM, we use a
binary score $R \in \{0, 1\}$, which indicates whether the LLM output is the (single) correct ($R = 1$) or
incorrect ($R = 0$) answer of the given options. To obtain reliable conclusions, we experiment with
each multiple choice question $10$ times, each time using a (different) random seed to generate the
Gumbel noise variables used by the sampler. Due to space constraints, in what follows, we compare
`1B` and `3B` on the knowledge area "college computer science". In Appendix E, we provide further
results on other knowledge areas and other pairs of LLMs.

**Results.** Figures 2a and 2b show that the scores of the LLMs are positively correlated under coupled
generation and thus the variance of the difference in scores is lower under coupled generation than
under independent, in agreement with Proposition 3.1. Further, we compute the error in the estimation
of the expected difference in scores resulting from using the two approaches as a function of the
available sample size. To this end, we first estimate the expected score difference using $1,000$ samples
and consider this as (a proxy of) the ground truth. Then, we compute the absolute estimation error
achieved by independent and coupled generation while sub-sampling the original samples across
various sample sizes. Figure 2c summarizes the results, which show that, as expected from our
theoretical analysis, a lower variance of the difference in scores under coupled generation leads to
a reduction in the number of samples required to achieve equivalent error in the estimation of the

| LLM | Coupled | | Independent | |
|---|---|---|---|---|
| | Rank | Avg. win-rate | Rank | Avg. win-rate |
| 8B | 1 | $0.3670 \pm 0.0020$ | 1 | $0.3863 \pm 0.0020$ |
| bnb-8bit | 2 | $0.3562 \pm 0.0020$ | 1 | $0.3825 \pm 0.0020$ |
| bnb-4bit | 3 | $0.3339 \pm 0.0020$ | 3 | $0.3463 \pm 0.0020$ |
| AWQ-INT4 | 4 | $0.3164 \pm 0.0019$ | 4 | $0.3310 \pm 0.0019$ |
| 3B | 5 | $0.2787 \pm 0.0019$ | 5 | $0.2828 \pm 0.0019$ |
| 1B | 6 | $0.1650 \pm 0.0015$ | 6 | $0.1664 \pm 0.0015$ |

Table 1: **Average win-rate and ranking of each LLM on questions from the LMSYS-Chat-1M dataset.** To estimate the average win-rate of each LLM, along with $95\%$ confidence intervals, we use the pairwise comparisons between the outputs of all pairs of LLMs using all $500$ questions with $10$ (different) random seeds under both coupled and independent generation. To derive the rankings, for each LLM, we choose the lowest ranking provided by the method of Chatzi et al. (2024b).

expected difference between the scores of the LLMs. Perhaps surprisingly, we find that this reduction can, in practice, be quite large. For example, to achieve an estimation error of $\approx 0.034$, coupled generation needs $40\%$ fewer samples than independent generation.

## 4.2 EVALUATION ON THE LMSYS-CHAT-1M DATASET

In this section, we compare the same three LLMs as in the previous section as well as three quantized variants, namely, `Llama-3.1-8B-Instruct-{AWQ-INT4, bnb-4bit, bnb-8bit}` (refer to Appendix D for more details), using pairwise comparisons between their outputs by a strong LLM, when prompted with open-ended questions from the LMSYS Chatbot Arena platform LMSYS (2023). Here, our goal is to investigate to what extent the win-rates under coupled and independent autoregressive generation differ in practice—our theoretical results in Appendix B show that such discrepancies do arise in certain canonical settings.

**Experimental setup.** We experiment with $500$ questions from the LMSYS-Chat-1M dataset Zheng et al. (2024), we provide the question itself as an input prompt to the LLMs, and instruct them to generate a concise response as an output through a system prompt. Further, similarly as elsewhere Chatzi et al. (2024b); Li et al. (2024a); Boyeau et al. (2024); Gera et al. (2024); Li et al. (2024b); Zheng et al. (2023), we use a strong LLM, namely, `GPT-4o-2024-11-20`, as a judge. More specifically, for each question and pair of outputs provided by two different LLMs, we prompt the judge to respond which of the two outputs it prefers, but allowing the judge to declare a tie (for the exact prompts we use, refer to Appendix D). Given these pairwise comparisons, to evaluate the outputs provided by each LLM, we use the win-rate achieved by each LLM against each other. To obtain reliable conclusions, similarly as in the previous section, we repeat each experiment $10$ times, each time using a (different) random seed to generate the Gumbel noise variables used by the Gumbel-Max SCM.

**Results.** We find that the empirical win-rate of each LLM against any other LLM is generally lower under coupled generation than under independent generation, as shown in Figure 3 for `8B`, `bnb-8bit` and `bnb-4bit`, and Figure 6 in Appendix F for other LLMs. Moreover, whenever the LLMs under comparison are *sufficiently* similar, the difference between win-rates is statistically significant, suggesting that our theoretical results may generalize beyond the canonical setting discussed in Appendix B. We hypothesize that this is partially due to an increase in the number of ties under coupled autoregressive generation. For example, for `bnb-8bit`, we observe a $24\%$, $11\%$, $15\%$ increase in the number of ties in the pairwise comparisons against `8B`, `bnb-4bit`, and `AWQ-INT4`. Remarkably, the difference in empirical win-rates leads to differences in the rankings derived from the average win-rates, as shown in Table 1. Under independent generation, the average win-rates achieved by `8B` and `bnb-8bit` are statistically indistinguishable and thus both are ranked at the top. However, under coupled generation, `8B` has a competitive advantage against `bnb-8bit`, and it is ranked at the top.

## 5 DISCUSSION AND LIMITATIONS

In this section, we discuss several aspects of our work, which we believe are important to consider and may serve as a basis for future research.

**Model assumptions.** Our theoretical analysis of coupled autoregressive generation focuses on sampling mechanisms that satisfy counterfactual stability (Oberst & Sontag, 2019). Although counterfactual stability has been shown to be a desirable property for causal mechanisms in SCMs and, more specifically, for causal mechanisms used for sampling in LLMs (Chatzi et al., 2024a), counterfactual stability may not always be appropriate and should be justified by domain specific knowledge (Haugh & Singal, 2023). In this context, it is also worth mentioning that the Gumbel-Max SCM is not the only SCM that satisfies counterfactual stability (Lorberbom et al., 2021; Haugh & Singal, 2023). Therefore, it would be interesting to understand the sensitivity of coupled autoregressive generation to this specific choice of SCM as well as extending our theoretical analysis to sampling mechanisms satisfying other alternative properties (Vlontzos et al., 2023).

**Practical considerations.** Our experimental results and theoretical analysis suggest that coupled autoregressive generation is most advantageous over independent autoregressive generation whenever the LLMs under comparison are sufficiently close in terms of their next-token distributions. Motivated by this observation, it would be important to identify which parts of the LLM development pipeline (*e.g.*, the LLMs' architectures, training data, or fine-tuning process) lead, in practice, to sufficiently small changes in the next-token distributions for coupled autoregressive generation to be most beneficial.

Our causal model for coupled autoregressive generation assumes that the LLMs under comparison share the same vocabulary. However, in practice, this may not hold since models use different tokenizers—different families of tokenizers may even use different low-level representations for tokens that appear to be the same at the string level.[11] One could think of naively lifting this assumption by merging the vocabularies of different LLMs, however, we empirically found that, using this strategy, different LLMs end up using different tokens (and thus noise values) to generate the same responses and thus coupled autoregressive generation provides significantly lower gains. Extending our causal model for coupled autoregressive generation to LLMs with different tokenizers is an interesting, albeit challenging, direction for future work.

**Evaluation.** We have conducted experiments using LLMs from the `Llama` family, namely `Llama-3.1-8B-Instruct` and `Llama-3.2-{1B, 3B}-Instruct`, and quantized versions thereof. It would be interesting to conduct experiments with LLMs from other families and also consider fine-tuned versions of them to understand how coupled autoregressive generation behaves in different settings. Furthermore, we have experimented with (i) a single benchmark dataset (*i.e.*, MMLU) and (ii) a single dataset of prompts for pairwise comparisons (*i.e.*, LMSYS Chatbot Arena), where we have used a strong LLM as a judge (*i.e.*, `GPT-4o-2024-11-20`) and win-rate as an evaluation metric. To better understand the benefits of coupled autoregressive generation, it would be important to experiment with additional datasets, pairwise comparisons made by humans, and additional evaluation metrics based on, *e.g.*, the Elo rating system Askell et al. (2021); Dettmers et al. (2024); Bai et al. (2022); Wu et al. (2023); Lin & Chen (2023) and the Bradley-Terry model Chiang et al. (2025); Boyeau et al. (2024).

## 6 CONCLUSIONS

In this work, we have introduced a causal model of coupled autoregressive generation that enables the evaluation and comparison of different LLMs under the same source of randomness. In several canonical settings, we have shown that, in evaluations based on benchmark datasets, coupled autoregressive generation can provably reduce the number of samples required to reliably compare the performance of LLMs and, in evaluations based on pairwise comparisons, it can provably lead to different and, perhaps more intuitive, rankings of LLMs in comparison with independent autoregressive generation. Lastly, we have empirically demonstrated that our theoretical results generalize to several state-of-the-art LLMs and datasets commonly used for the evaluation and ranking of LLMs.

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

## A  FORMAL DEFINITION OF COUNTERFACTUAL STABILITY

Counterfactual stability is a desirable property of SCMs (Oberst & Sontag, 2019) that has previously been used in the context of autoregressive generation of LLMs Chatzi et al. (2024a). In the following, we provide its formal definition along with a simple example to explain the intuition behind it. Throughout this section, $P^{\mathcal{C}\,;\,do(\cdot)}$ denotes the probability of the interventional distribution entailed by an SCM $\mathcal{C}$ under an intervention $do(\cdot)$. Moreover, $P^{\mathcal{C}\,|\,\star\,;\,do(\cdot)}$ denotes the probability of the counterfactual distribution entailed by an SCM $\mathcal{C}$ under an intervention $do(\cdot)$ given that an observed event $\star$ has already occurred.

**Definition A.1.** A sampling mechanism defined by $f_T$ and $P_U$ satisfies counterfactual stability if for all LLMs $m, m' \in \mathcal{M}$, $i \in \{1, 2, \ldots, K\}$ and tokens $t_1, t_2 \in V$ with $t_1 \neq t_2$, the condition

$$\frac{P^{\mathcal{C}\,;\,do(M=m')}[T_i = t_1 \mid D_i]}{P^{\mathcal{C}\,;\,do(M=m)}[T_i = t_1 \mid D_i]} \geq \frac{P^{\mathcal{C}\,;\,do(M=m')}[T_i = t_2 \mid D_i]}{P^{\mathcal{C}\,;\,do(M=m)}[T_i = t_2 \mid D_i]} \tag{6}$$

implies that $P^{\mathcal{C}\,|\,D_i, M=m, T_i=t_1\,;\,do(M=m')}[T_i = t_2] = 0$.

The property of counterfactual stability has an intuitive interpretation that can be best understood via a simple example. Assume that the vocabulary contains 2 tokens "A" and "B" and, using LLM $m$, the next-token distribution at a time step $i$ assigns values $0.6$, $0.4$ to the two tokens, respectively. Moreover, the realized noise value $\mathbf{u}_i$ is such that the token "A" is sampled. Now, consider that, while keeping the noise value $\mathbf{u}_i$ fixed, we change the LLM to $m'$, resulting in a next-token distribution that assigns values $0.7$, $0.3$ to the two tokens, respectively. Counterfactual stability ensures that, since the noise value $\mathbf{u}_i$ led to "A" being sampled under $m$ at $0.6$ to $0.4$ odds, the same value cannot lead to "B" being sampled under $m'$ where its relative odds are lower (*i.e.*, $0.3$ to $0.7$).

## B  THEORETICAL ANALYSIS OF EVALUATIONS BASED ON PAIRWISE COMPARISONS

In this section, we focus on the evaluation and comparison of LLMs according to their level of alignment with human preferences, as elicited by pairwise comparisons between outputs of different LLMs to the same prompts. Such an evaluation protocol has become particularly popular to evaluate and compare LLMs in open-ended, complex tasks in which, in contrast to benchmark datasets, there are no structured ground-truth outputs. In what follows, we provably show that, perhaps surprisingly, different LLMs may compare differently under coupled autoregressive generation and under independent autoregressive generation.

A standard approach to evaluate and compare different LLMs according to their level of alignment with human pairwise preferences reduces to estimating the win-rate achieved by each LLM $m$ against

any other LLM $m' \neq m$, *i.e.*,[12]

$$\mathbb{E}_{\boldsymbol{U}, \boldsymbol{U}' \sim P_{\boldsymbol{U}}, S_q \sim P_Q}[\mathbf{1}\{R_m(\boldsymbol{U}, S_q) > R_{m'}(\boldsymbol{U}', S_q)\}] \tag{7}$$

Independent generation

where $\mathbf{1}\{R_m(\mathbf{u}, s_q) > R_{m'}(\mathbf{u}, s_q)\} = 1\,(0)$ means that, for prompt $s_q$ and realized sequence of noise values $\mathbf{u}$, the output of $m$ is (not) preferred over the output of $m'$.[13]

Here, similarly as in Eq. 2 in the evaluation based on benchmark datasets, we use different noise variables $\boldsymbol{U}$ and $\boldsymbol{U}'$ because, in this standard approach, each LLM generates outputs to each prompt independently (*i.e.*, using independent generation). Conversely, under coupled autoregressive generation, the win-rate adopts the following form:

$$\mathbb{E}_{\boldsymbol{U} \sim P_{\boldsymbol{U}}, S_q \sim P_Q}[\mathbf{1}\{R_m(\boldsymbol{U}, S_q) > R_{m'}(\boldsymbol{U}, S_q)\}] \tag{8}$$

Coupled generation

However, in contrast with the comparison of the expected difference in scores under independent and coupled autoregressive generation in the evaluation based on benchmark datasets, we cannot directly claim that Eqs. 7 and 8 are equivalent because the win-rate is non-linear with respect to $R_m(\mathbf{u}, s_q)$ and $R_{m'}(\mathbf{u}', s_q)$. In what follows, we will further analyze the difference between win-rates in two canonical settings similar to those we used in Section 3.

In the first canonical setting, for each prompt, the response can only be one of two given single-token sequences and one of these sequences is preferred over the other by the user. Further, the LLMs under comparison always output one of them as a response and the sampling mechanism used by the LLMs satisfies counterfactual stability. Then, we can compute the win-rates achieved by each LLM $m$ against any other LLM $m' \neq m$ under independent and coupled autoregressive generation using the following proposition:

**Proposition B.1.** *Given a fixed prompt $s_q \sim P_Q$, assume that $f_R(s_+) > f_R(s_-)$ for $s_+ = s_q \circ t_+$ and $s_- = s_q \circ t_-$, where $t_+$ and $t_-$ are single-token sequences. Further, assume that the LLMs $m$ and $m'$ respond $t_+$ with probability $p_m$ and $p_{m'}$, respectively, and $t_-$ with probability $1 - p_m$ and $1 - p_{m'}$, and the sampling mechanism defined by $f_T$ and $P_U$ satisfies counterfactual stability. Without loss of generality, assume $p_{m'} > p_m$. Then, under coupled autoregressive generation, we have that*

$$\mathbb{E}_{\boldsymbol{U} \sim P_{\boldsymbol{U}}}[\mathbf{1}\{R_m(\boldsymbol{U}, s_q) > R_{m'}(\boldsymbol{U}, s_q)\}] = 0,$$
$$\mathbb{E}_{\boldsymbol{U} \sim P_{\boldsymbol{U}}}[\mathbf{1}\{R_m(\boldsymbol{U}, s_q) < R_{m'}(\boldsymbol{U}, s_q)\}] = p_{m'} - p_m.$$

*Conversely, under independent autoregressive generation, we have that*

$$\mathbb{E}_{\boldsymbol{U}, \boldsymbol{U}' \sim P_{\boldsymbol{U}}}[\mathbf{1}\{R_m(\boldsymbol{U}, s_q) > R_{m'}(\boldsymbol{U}', s_q)\}] = p_m(1 - p_{m'}),$$
$$\mathbb{E}_{\boldsymbol{U}, \boldsymbol{U}' \sim P_{\boldsymbol{U}}}[\mathbf{1}\{R_m(\boldsymbol{U}, s_q) < R_{m'}(\boldsymbol{U}', s_q)\}] = p_{m'}(1 - p_m)$$

From the above proposition, we can readily conclude that, in general, the win-rates do differ under independent and coupled autoregressive generation. Nevertheless, we may be tempted to conclude that, for ranking LLMs, this difference appears inconsequential because, for each fixed prompt $s_q$, we have that the difference of win-rates match. However, whenever one needs to rank more than two LLMs, the difference in win-rates can be actually consequential—the rankings derived from the win-rates can be different under independent and coupled autoregressive generation, as illustrated by the following simple example.

Consider we are given three LLMs $m_1$, $m_2$, and $m_3$, and we need to rank them according to the average win-rate they achieve against each other on two input prompts $q$ and $q'$, each with a preferred single-token response out of two single-token responses. Assume that the probability that each LLM outputs the preferred single-token response for $q$ and $q'$ is given by the table of the example introduced in Section 1. Under independent autoregressive generation, the average win-rates of $m_1$, $m_2$ and $m_3$

---

[12]We believe that our theoretical results can be extended to other popular performance metrics based on the Elo rating system Askell et al. (2021); Dettmers et al. (2024); Bai et al. (2022); Wu et al. (2023); Lin & Chen (2023) and the Bradley-Terry model Chiang et al. (2025); Boyeau et al. (2024), as discussed in Section 5.

[13]For simplicity, we assume that human preferences are deterministic and thus $R_m(\mathbf{u}, s_q, z) = R_m(\mathbf{u}, s_q)$. We lift this assumption in Section 4.

are $0.1545$, $0.15675$ and $0.16225$, respectively. Therefore, $m_3$ is ranked at the top, followed by $m_2$, and $m_1$ is ranked last. In contrast, under coupled autoregressive generation, the average win-rates of $m_1$, $m_2$ and $m_3$ are $0.0525$, $0.0225$, and $0.03$, respectively (refer to Appendix C.6 for the detailed calculation), and thus $m_1$ is ranked at the top, followed by $m_3$, and $m_2$ is ranked last. Interestingly, the ranking obtained under coupled autoregressive generation aligns with the ranking obtained in Section 1 using the average accuracy of each LLM. More crucially, this case illustrates how rankings obtained using coupled and independent autoregressive generation can differ, leading to opposite conclusions regarding the LLMs' performance.

In the second canonical setting, for each prompt, the response can be one of any single-token sequences, and each of the sequences may provide a different level of user's satisfaction (*i.e.*, achieve a different score). Further, the LLMs under comparison always output one of them as a response and the sampling mechanism used by the LLMs is given by the Gumbel-Max SCM. The following proposition shows that the number of ties between an LLM $m$ and any other *sufficiently similar* LLM $m' \neq m$ are higher under coupled autoregressive generation than under independent autoregressive generation:

**Proposition B.2.** *Given a fixed prompt $s_q \sim P_Q$, assume, without loss of generality, that $f_R(s_q \circ t_1) \geq f_R(s_q \circ t_2) \geq \ldots \geq f_R(s_q \circ t_{|V|})$. Let $m$ be an LLM that assigns positive probability to every single-token sequence and zero probability to any other sequence. If the sampling mechanism defined by $f_T$ and $P_U$ is given by the Gumbel-Max SCM, then, there exists a constant $\varepsilon(m) > 0$ such that, for every LLM $m'$ that assigns positive probability to every single-token sequence and zero probability to any other sequence and satisfies $d(m, m') = \sup_{s_q} \|f_D(s_q, m) - f_D(s_q, m')\|_\infty < \varepsilon(m)$, it holds that*

$$\mathbb{E}_{\boldsymbol{U} \sim P_U}[\mathbf{1}\{R_m(\boldsymbol{U}, s_q) = R_{m'}(\boldsymbol{U}, s_q)\}] > \mathbb{E}_{\boldsymbol{U}, \boldsymbol{U}' \sim P_U}[\mathbf{1}\{R_m(\boldsymbol{U}, s_q) = R_{m'}(\boldsymbol{U}', s_q)\}].$$

The above proposition implies that the win-rates under independent and coupled autoregressive generation are different and, similarly as in the first canonical setting, rankings derived from the win-rates may differ under independent and coupled autoregressive generation. We investigate this further in our experiments in Section 4.

## C PROOFS

### C.1 PROOF OF PROPOSITION 3.1

We can rewrite the variance of the difference in scores under independent generation in terms of the variance of the difference in scores under coupled generation as follows:

$$\begin{aligned}
&\mathrm{Var}[R_m(\boldsymbol{U}, S_q) - R_{m'}(\boldsymbol{U}', S_q)]] \\
&= \mathrm{Var}[R_m(\boldsymbol{U}, S_q) - R_{m'}(\boldsymbol{U}, S_q) + R_{m'}(\boldsymbol{U}, S_q) - R_{m'}(\boldsymbol{U}', S_q)]] \\
&= \mathrm{Var}[R_m(\boldsymbol{U}, S_q) - R_{m'}(\boldsymbol{U}, S_q)] + \mathrm{Var}[R_{m'}(\boldsymbol{U}, S_q) - R_{m'}(\boldsymbol{U}', S_q)] \\
&\quad + 2 \cdot \mathrm{Cov}[R_m(\boldsymbol{U}, S_q) - R_{m'}(\boldsymbol{U}, S_q), R_{m'}(\boldsymbol{U}, S_q) - R_{m'}(\boldsymbol{U}', S_q)].
\end{aligned}$$

For the variance of the difference in scores for the same LLM under independent noise values, we have that

$$\begin{aligned}
&\mathrm{Var}[R_{m'}(\boldsymbol{U}, S_q) - R_{m'}(\boldsymbol{U}', S_q)] \\
&\stackrel{(a)}{=} \mathbb{E}[(R_{m'}(\boldsymbol{U}, S_q) - R_{m'}(\boldsymbol{U}', S_q))^2] - \mathbb{E}[R_{m'}(\boldsymbol{U}, S_q) - R_{m'}(\boldsymbol{U}', S_q)]^2 \\
&\stackrel{(b)}{=} \mathbb{E}[R_{m'}(\boldsymbol{U}, S_q)^2 - 2 \cdot R_{m'}(\boldsymbol{U}, S_q)R_{m'}(\boldsymbol{U}', S_q) + R_{m'}(\boldsymbol{U}', S_q)^2] \\
&\stackrel{(c)}{=} 2 \cdot \mathbb{E}[R_{m'}(\boldsymbol{U}, S_q)^2] - 2 \cdot \mathbb{E}[R_{m'}(\boldsymbol{U}, S_q)R_{m'}(\boldsymbol{U}', S_q)],
\end{aligned}$$

where (a) holds by the definition of variance, (b) is due to the subtraction term being $0$, and (c) is due to the linearity of expectation. Further, for the covariance of the difference in scores under independent generation and the difference in scores under coupled generation, we have that

$$\begin{aligned}
&\mathrm{Cov}[R_m(\boldsymbol{U}, S_q) - R_{m'}(\boldsymbol{U}, S_q), R_{m'}(\boldsymbol{U}, S_q) - R_{m'}(\boldsymbol{U}', S_q)] \\
&\stackrel{(a)}{=} \mathbb{E}[(R_m(\boldsymbol{U}, S_q) - R_{m'}(\boldsymbol{U}, S_q)) \cdot (R_{m'}(\boldsymbol{U}, S_q) - R_{m'}(\boldsymbol{U}', S_q))]
\end{aligned}$$

$$- \mathbb{E}[R_m(\boldsymbol{U}, S_q) - R_{m'}(\boldsymbol{U}, S_q)] \cdot \mathbb{E}[R_{m'}(\boldsymbol{U}, S_q) - R_{m'}(\boldsymbol{U}', S_q)]$$

$$\overset{(b)}{=} \mathbb{E}[R_m(\boldsymbol{U}, S_q) R_{m'}(\boldsymbol{U}, S_q)] - \mathbb{E}[R_m(\boldsymbol{U}, S_q) R_{m'}(\boldsymbol{U}', S_q)]$$
$$- \mathbb{E}[R_{m'}(\boldsymbol{U}, S_q) R_{m'}(\boldsymbol{U}, S_q)] + \mathbb{E}[R_{m'}(\boldsymbol{U}, S_q) R_{m'}(\boldsymbol{U}', S_q)]$$

$$\overset{(c)}{=} \mathrm{Cov}[R_m(\boldsymbol{U}, S_q), R_{m'}(\boldsymbol{U}, S_q)] - \mathbb{E}[R_{m'}(\boldsymbol{U}, S_q)^2] + \mathbb{E}[R_{m'}(\boldsymbol{U}, S_q) R_{m'}(\boldsymbol{U}', S_q)]]$$

where (a) and (c) hold by the definition of covariance and (b) is due to the last term being zero and by the expansion of the first term.

Putting all the above results together, it follows that

$$\mathrm{Var}[R_m(\boldsymbol{U}, S_q) - R_{m'}(\boldsymbol{U}', S_q)]]$$
$$= \mathrm{Var}[R_m(\boldsymbol{U}, S_q) - R_{m'}(\boldsymbol{U}, S_q)] + 2 \cdot \mathrm{Cov}\left[R_m(\boldsymbol{U}, S_q), R_{m'}(\boldsymbol{U}, S_q)\right]$$
$$+ 2 \cdot \mathbb{E}[R_{m'}(\boldsymbol{U}, S_q) R_{m'}(\boldsymbol{U}', S_q)] - 2 \cdot \mathbb{E}[R_{m'}(\boldsymbol{U}, S_q)^2] + 2 \cdot \mathbb{E}[R_{m'}(\boldsymbol{U}, S_q)^2]$$
$$- 2 \cdot \mathbb{E}[R_{m'}(\boldsymbol{U}, S_q) R_{m'}(\boldsymbol{U}', S_q)]$$
$$= \mathrm{Var}[R_m(\boldsymbol{U}, S_q) - R_{m'}(\boldsymbol{U}, S_q)] + 2 \cdot \mathrm{Cov}[R_m(\boldsymbol{U}, S_q), R_{m'}(\boldsymbol{U}, S_q)]$$

which concludes the proof.

## C.2 PROOF OF PROPOSITION 3.2

Due to Proposition 3.1, to show that Eq. 5 holds, it suffices to show that the covariance between the scores of the different LLMs under coupled generation is non-negative, *i.e.*, $\mathrm{Cov}[R_m(\boldsymbol{U}, S_q), R_{m'}(\boldsymbol{U}, S_q)] \geq 0$.

To this end, we first rewrite the covariance as

$$\mathrm{Cov}[R_m(\boldsymbol{U}, S_q), R_{m'}(\boldsymbol{U}, S_q)]$$
$$= P[R_m(\boldsymbol{U}, S_q) = 1, R_{m'}(\boldsymbol{U}, S_q) = 1] - P[R_m(\boldsymbol{U}, S_q) = 1] \cdot P[R_{m'}(\boldsymbol{U}, S_q) = 1]$$
$$= \sum_{s_q} P[S_q = s_q] \cdot (P[R_m(\boldsymbol{U}, s_q) = 1, R_{m'}(\boldsymbol{U}, s_q) = 1] \tag{9}$$
$$- P[R_m(\boldsymbol{U}, s_q) = 1] \cdot P[R_{m'}(\boldsymbol{U}, s_q) = 1])$$

Next, we note that the event $R_m(\boldsymbol{U}, s_q) = 1$ is equivalent to LLM $m$ sampling the ground truth token for prompt $s_q$. Without loss of generality, assume $t_1$ is the ground truth token, *i.e.*, $\mathrm{C}(s_q) = t_1$. Then, since only tokens $\{t_1, t_2\}$ have positive probability under $m$ and $m'$, it must hold that either (i) one LLM assigns a greater probability to $t_1$ and the other LLM assigns a greater probability to $t_2$, or (ii) both LLMs assign the same probabilities. Further, since the sampling mechanism defined by $f_T$ and $P_U$ satisfies counterfactual stability, we have that the condition in Eq. 6 holds in both (i) and (ii) and, under coupled generation, the LLM with greater (or equal) probability for $t_1$ will always sample $t_1$ when the LLM with lower (or equal) probability does. This implies that

$$P[R_m(\boldsymbol{U}, s_q) = 1, R_{m'}(\boldsymbol{U}, s_q) = 1] = \min\{P[R_m(\boldsymbol{U}, s_q) = 1], P[R_{m'}(\boldsymbol{U}, s_q) = 1]\} \tag{10}$$

Finally, since it holds that

$$\min\{P[R_m(\boldsymbol{U}, s_q) = 1], P[R_{m'}(\boldsymbol{U}, s_q) = 1]\} \geq P[R_m(\boldsymbol{U}, s_q) = 1] P[R_{m'}(\boldsymbol{U}, s_q) = 1] \tag{11}$$

because $P[R_m(\boldsymbol{U}, s_q) = 1] \in (0, 1)$ and $P[R_{m'}(\boldsymbol{U}, s_q) = 1] \in (0, 1)$ by assumption, we can conclude from Eq. 9 that

$$\mathrm{Cov}[R_m(\boldsymbol{U}, S_q), R_{m'}(\boldsymbol{U}, S_q)] > 0. \tag{12}$$

## C.3 PROOF OF PROPOSITION 3.3

Using Proposition 3.1, we have that

$$\mathrm{Cov}[R_m(\boldsymbol{U}, S_q), R_{m'}(\boldsymbol{U}, S_q)]$$
$$= \mathbb{E}[R_m(\boldsymbol{U}, S_q) \cdot R_{m'}(\boldsymbol{U}, S_q)] - \mathbb{E}[R_m(\boldsymbol{U}, S_q)] \cdot \mathbb{E}[R_{m'}(\boldsymbol{U}, S_q)]$$
$$= \underbrace{P[R_m(\boldsymbol{U}, S_q) = 1, R_{m'}(\boldsymbol{U}, S_q) = 1]}_{(i)} - \underbrace{P[R_m(\boldsymbol{U}, S_q) = 1) \cdot P[R_{m'}(\boldsymbol{U}, S_q) = 1]}_{(ii)}.$$

In the remainder of the proof, we will bound each term (i) and (ii) separately and, since $|\mathsf{C}(s_q)| = 1$ for all $s_q \sim P_Q$, assume without loss of generality that the correct token is single-token sequence $t_1$.

To bound the term (ii), first note that, using the definition of the Gumbel-Max SCM, we have that, for each $k \in \{2, \ldots, |V|\}$, it holds that

$$R_m(\boldsymbol{U}, s_q) = 1 \iff U_1 + \log([f_D(s_q, m)]_{t_1}) \geq U_k + \log([f_D(s_q, m)]_{t_k}),$$
$$R_{m'}(\boldsymbol{U}, s_q) = 1 \iff U_1 + \log([f_D(s_q, m')]_{t_1}) \geq U_k + \log([f_D(s_q, m')]_{t_k}).$$

Next, let $\varepsilon^* > 0$ be an arbitrary constant that we will determine later such that

$$|\log([f_D(S_q, m)]_{t_k}) - \log([f_D(S_q, m')]_{t_k})| \leq \varepsilon^*, \tag{13}$$

and note that since, by assumption, $D_{t_k} > 0$ for all $k \in \{1, \ldots, |V|\}$, any bound on the absolute difference of log-probabilities $|\log([f_D(S_q, m)]_{t_k}) - \log([f_D(S_q, m')]_{t_k})|$ uniformly implies a bound on the difference of probabilities $|[f_D(S_q, m)]_{t_k} - [f_D(S_q, m')]_{t_k}|$ and vice versa. For simplicity, we prove the result in the log-domain.

Now, using the bound defined by Eq. 13, we have that

$$\bigcap_{k \neq 1} \{U_1 + \log([f_D(S_q, m')]_{t_1}) \geq U_k + \log([f_D(S_q, m')]_{t_k})\}$$

$$\subset \bigcap_{k \neq 1} \{U_1 + \log([f_D(S_q, m)]_{t_1}) + \varepsilon^* \geq U_k + \log([f_D(S_q, m)]_{t_k}) - \varepsilon^*\},$$

and we can then bound the term (ii) as follows:

$$P[R_m(\boldsymbol{U}, S_q) = 1] \cdot P[R_{m'}(\boldsymbol{U}, S_q) = 1]$$
$$= P[\cap_{k \neq 1}\{U_1 + \log([f_D(S_q, m)]_{t_1}) \geq U_k + \log([f_D(S_q, m)]_{t_k})\}]$$
$$\times P[\cap_{k \neq 1}\{U_1 + \log([f_D(S_q, m')]_{t_1}) \geq U_k + \log([f_D(S_q, m')]_{t_k})\}]$$
$$\leq P[\cap_{k \neq 1}\{U_1 + \log([f_D(S_q, m)]_{t_1}) \geq U_k + \log([f_D(S_q, m)]_{t_k})\}]$$
$$\times P[\cap_{k \neq 1}\{U_1 + \log([f_D(S_q, m)]_{t_1}) + \varepsilon^* \geq U_k + \log([f_D(S_q, m)]_{t_k}) - \varepsilon^*\}].$$

To bound the term (i), first note that, using the bound defined by Eq. 13, we have that

$$\bigcap_{k \neq 1} \{U_1 + \log([f_D(S_q, m')]_{t_1}) \geq U_k + \log([f_D(S_q, m')]_{t_k})\}$$

$$\supset \bigcap_{k \neq 1} \{U_1 + \log([f_D(S_q, m)]_{t_1}) - \varepsilon^* \geq U_k + \log([f_D(S_q, m)]_{t_k}) + \varepsilon^*\}.$$

Thus, we can bound the term (i) as follows:

$$P[R_m(\boldsymbol{U}, S_q) = 1, R_{m'}(\boldsymbol{U}, S_q) = 1]$$
$$= P\Big[\cap_{k \neq 1} \{U_1 + \log([f_D(S_q, m)]_{t_1}) \geq U_k + \log([f_D(S_q, m)]_{t_k})\}$$
$$\cap \{U_1 + \log([f_D(S_q, m')]_{t_1}) \geq U_k + \log([f_D(S_q, m')]_{t_k})\}\Big]$$
$$\geq P\Big[\cap_{k \neq 1} \{U_1 + \log([f_D(S_q, m)]_{t_1}) \geq U_k + \log([f_D(S_q, m)]_{t_k})\}$$
$$\cap \{U_1 + \log([f_D(S_q, m)]_{t_1}) \geq U_k + \log([f_D(S_q, m)]_{t_k}) + 2\varepsilon^*\}\Big]$$
$$\overset{(a)}{=} \sum_{s_q} P[S_q = s_q] \cdot P[\cap_{k \neq 1}\{U_1 + \log([f_D(S_q, m)]_{t_1})$$
$$\geq U_k + \log([f_D(S_q, m)]_{t_k}) + 2\varepsilon^*\}],$$

where (a) follows from the fact that

$$\{U_1 + \log([f_D(S_q, m)]_{t_1}) \geq U_k + \log([f_D(S_q, m)]_{t_k}) + 2\varepsilon^*\}$$
$$\subset \{U_1 + \log([f_D(S_q, m)]_{t_1}) \geq U_k + \log([f_D(S_q, m)]_{t_k})\}.$$

Now, note that, for $k \in \{2, \ldots, |V|\}$, the variable $X_k \equiv U_1 - U_k \sim \text{Logistic}(0, 1)$ (for $k = 1$, define $X_k \equiv 0$). Therefore, we can rewrite the bound for (i) as

$$P[R_m(\boldsymbol{U}, S_q) = 1, R_{m'}(\boldsymbol{U}, S_q) = 1]$$
$$\geq \sum_{s_q} P[S_q = s_q] \cdot \prod_{k \neq 1} \cdot P[\{X_k \geq \log([f_D(S_q, m)]_{t_k}) - \log([f_D(S_q, m)]_{t_1}) + 2\varepsilon^*\}]$$

and we can rewrite the bound for (ii) as

$$P[R_m(\boldsymbol{U}, S_q) = 1]P[R_{m'}(\boldsymbol{U}, S_q) = 1] \leq$$
$$\sum_{s_q} P[S_q = s_q] \cdot \left\{ \prod_{k \neq 1} P[\{X_k \geq \log([f_D(S_q, m)]_{t_k}) - \log([f_D(S_q, m)]_{t_k}) - 2\varepsilon^*\}] \right\}$$
$$\times P[\cap_{k \neq 1}\{X_k \geq \log([f_D(S_q, m)]_{t_k}) - \log([f_D(S_q, m)]_{t_k})\}].$$

As a consequence, to prove that $P[R_m(\boldsymbol{U}, S_q) = 1, R_{m'}(\boldsymbol{U}, S_q) = 1] > P[R_m(\boldsymbol{U}, S_q) = 1]P[R_{m'}(\boldsymbol{U}, S_q) = 1]$, it suffices to show that

$$\sum_{s_q} P[S_q = s_q] \prod_{k \neq 1} \cdot P[\{X_k \geq \log([f_D(S_q, m)]_{t_k}) - \log([f_D(S_q, m)]_{t_1}) + 2\varepsilon^*\}]$$
$$> \sum_{s_q} P[S_q = s_q] \prod_{k \neq 1} \cdot P[\{X_k \geq \log([f_D(S_q, m)]_{t_k}) - \log([f_D(S_q, m)]_{t_1}) - 2\varepsilon^*\}]$$
$$\times P[\cap_{k \neq 1}\{X_k \geq \log([f_D(S_q, m)]_{t_k}) - \log([f_D(S_q, m)]_{t_1})\}] \quad (14)$$

To do so, note that Eq. 14 holds trivially for $\varepsilon^* = 0$ since

$$P[\cap_{k \neq 1}\{X_k \geq \log([f_D(S_q, m)]_{t_k}) - \log([f_D(S_q, m)]_{t_1})\}] < 1,$$

which is a fixed term independent of $m'$. Since all terms in Eq. 14 are continuous in $\varepsilon^*$, there exists $\varepsilon^*(m) > 0$, possibly dependent of $m$ but independent of $m'$, such that Eq. 14 holds if

$$\sup_{s_q} \|\log(f_D(s_q, m)) - \log(f_D(s_q, m'))\|_\infty < \varepsilon^*(m).$$

Since by assumption $D_t > 0$ for all $t \in V$, there exists $\varepsilon(m) > 0$ in probability space such that Eq. 14 holds if

$$\sup_{s_q} \|f_D(s_q, m) - f_D(s_q, m')\|_\infty < \varepsilon(m).$$

This concludes the proof.

### C.4 PROOF OF PROPOSITION B.1

Under coupled autoregressive generation, if the LLM $m$ samples the preferred token $t_+$, then the LLM $m'$ must also sample $t_+$ because $t_+$ is more likely under $m'$ than under $m$ and the sampling mechanism defined by $f_T$ and $P_U$ satisfies counterfactual stability. This implies that the win-rate achieved by $m$ against $m'$ is

$$\mathbb{E}_{\boldsymbol{U} \sim P_U}[\mathbf{1}\{R_m(\boldsymbol{U}, s_q) > R_{m'}(\boldsymbol{U}, s_q)\}]$$
$$= P[f_T(f_D(s_q, m), \boldsymbol{U}) = t_+, f_T(f_D(s_q, m'), \boldsymbol{U}) = t_-] = 0 \quad (15)$$

and that

$$P[f_T(f_D(s_q, m), \boldsymbol{U}) = t_+, f_T(f_D(s_q, m'), \boldsymbol{U}) = t_+] = P[f_T(f_D(s_q, m), \boldsymbol{U}) = t_+] = p_m. \quad (16)$$

Using the same reasoning, if the LLM $m'$ samples the non-preferred token $t_-$, then, $m$ must also sample $t_-$ because $t_-$ is more likely under $m$ than under $m'$. This implies that

$$P[f_T(f_D(s_q, m), \boldsymbol{U}) = t_-, f_T(f_D(s_q, m'), \boldsymbol{U}) = t_-] = P[f_T(f_D(s_q, m'), \boldsymbol{U}) = t_-] = 1 - p_{m'} \quad (17)$$

Then, from Eq. 16 and Eq. 17, we can conclude that

$$\mathbb{E}_{\boldsymbol{U} \sim P_U}[\mathbf{1}\{R_m(\boldsymbol{U}, s_q) = R_{m'}(\boldsymbol{U}, s_q)\}] = p_m + (1 - p_{m'}) \quad (18)$$

Finally, from Eq. 15 and Eq. 18, we can conclude that the win-rate achieved by $m'$ against $m$ is

$$\mathbb{E}_{\boldsymbol{U} \sim P_{\boldsymbol{U}}}[\mathbf{1}\{R_m(\boldsymbol{U}, s_q) < R_{m'}(\boldsymbol{U}, s_q)\}]$$
$$= 1 - \mathbb{E}_{\boldsymbol{U} \sim P_{\boldsymbol{U}}}[\mathbf{1}\{R_m(\boldsymbol{U}, s_q) > R_{m'}(\boldsymbol{U}, s_q)\}] - \mathbb{E}_{\boldsymbol{U} \sim P_{\boldsymbol{U}}}[\mathbf{1}\{R_m(\boldsymbol{U}, s_q) = R_{m'}(\boldsymbol{U}, s_q)\}] = p_{m'} - p_m.$$

Under independent autoregressive generation, the LLMs $m$ and $m'$ sample tokens independently from each other, *i.e.*, $f_T(f_D(s_q, m), \boldsymbol{U}) \perp f_T(f_D(s_q, m'), \boldsymbol{U}')$. Thus, we can factorize all joint probabilities when computing the win-rates and obtain

$$\mathbb{E}_{\boldsymbol{U}, \boldsymbol{U}' \sim P_{\boldsymbol{U}}}[\mathbf{1}\{R_m(\boldsymbol{U}, s_q) > R_{m'}(\boldsymbol{U}', s_q)\}]$$
$$= P[f_T(f_D(s_q, m), \boldsymbol{U}) = t_+] \cdot P[f_T(f_D(s_q, m'), \boldsymbol{U}') = t_-] = p_m \cdot (1 - p_{m'})$$

and

$$\mathbb{E}_{\boldsymbol{U}, \boldsymbol{U}' \sim P_{\boldsymbol{U}}}[\mathbf{1}\{R_m(\boldsymbol{U}, s_q) < R_{m'}(\boldsymbol{U}', s_q)\}] = p_{m'} \cdot (1 - p_m).$$

## C.5 PROOF OF PROPOSITION B.2

We follow the notations and technique of Proposition 3.3. Fix query $s_q$ and consider first the case of independent autoregressive generation. Since each LLM can only assign a non-zero probability to single-token sequences, we have:

$$P[R_m(\boldsymbol{U}, s_q) = R_{m'}(\boldsymbol{U}', s_q)] = \sum_{k=1}^{|V|} P[f_T(f_D(s_q, m), \boldsymbol{U}) = t_k] P[f_T(f_D(s_q, m'), \boldsymbol{U}) = t_k]$$
$$< \sum_{k=1}^{|V|} P[f_T(f_D(s_q, m), \boldsymbol{U}) = t_k],$$

In the case of coupled autoregressive generation, since

$$P\left[\{f_T(f_D(s_q, m), \boldsymbol{U}) = t_k\} \cap \{f_T(f_D(s_q, m), \boldsymbol{U}) = t_j\}\right] = 0, \ i \neq j,$$

we obtain:

$$P[R_m(\boldsymbol{U}, s_q) = R_{m'}(\boldsymbol{U}, s_q)]$$
$$= P\left[\cup_i\{f_T(f_D(s_q, m), \boldsymbol{U}) = t_k, f_T(f_D(s_q, m'), \boldsymbol{U}) = t_k\}\right]$$
$$= \sum_k P[\{f_T(f_D(s_q, m), \boldsymbol{U}) = t_k, f_T(f_D(s_q, m'), \boldsymbol{U}) = t_k\}]$$
$$= \sum_k P[f_T(f_D(s_q, m), \boldsymbol{U}) = t_k] P[f_T(f_D(s_q, m'), \boldsymbol{U}) = t_k | f_T(f_D(s_q, m), \boldsymbol{U}) = t_k].$$

We now follow Huijben et al. (2023) and expand the posterior Gumbels, $P[f_T(f_D(s_q, m'), \boldsymbol{U}) = t_k | f_T(f_D(s_q, m), \boldsymbol{U}) = t_k]$, as truncated Gumbel distributions. In particular, we leverage the fact that

$$\max_{t \in V}\{U_t + \log([f_D(s_q, \bullet)]_t)\} \sim \text{Gumbel}(0, 1), \tag{19}$$

and that a Gumbel distribution, with parameter $\log(\theta)$, truncated at $b \sim \text{Gumbel}(0, 1)$ can be sampled as

$$-\log(\exp(-b) - \log(\eta)/\theta), \ \eta \sim U(0, 1). \tag{20}$$

Furthermore, by assumption, $D_{t_k} > 0$ for all $k \in \{1, \dots, |V|\}$, so that any bound on the absolute difference of log-probabilities $|\log([f_D(s_q, m)]_{t_k}) - \log([f_D(s_q, m')]_{t_k})|$ uniformly implies a bound on the difference of probabilities $|[f_D(s_q, m)]_{t_k} - [f_D(s_q, m')]_{t_k}|$ and vice versa. Using the bound

$$|\log([f_D(s_q, m)]_{t_k}) - \log([f_D(s_q, m')]_{t_k})| \leq \varepsilon^*$$

and the Gumbel properties in Eq. 19 and Eq. 20, we obtain:

$$P[R_m(\boldsymbol{U}, s_q) = R_{m'}(\boldsymbol{U}, s_q)]$$

$$= \sum_k P[f_T(f_D(s_q, m), \boldsymbol{U}) = t_k]$$

$$\times P\left[\bigcap_k \Big\{ \log([f_D(s_1, m')]_{t_k}) - \log([f_D(s_1, m)]_{t_k}) - \log(-\log(\eta_k)) \right.$$

$$\left. \geq \log([f_D(s_1, m')]_{t_j}) - \log([f_D(s_1, m)]_{t_j}) - \log(-\log(\eta_k) - \log(\eta_j)/[f_D(s_1, m')]_{t_j}) \Big\} \right]$$

$$\geq \sum_k P[f_T(f_D(s_q, m), \boldsymbol{U}) = t_k]$$

$$\times P\left[\cap_k \{-\log(-\log(\eta_k)) \geq -2\varepsilon^* - \log(-\log(\eta_k) - \log(\eta_j)/[f_D(s_1, m')]_{t_j})\}\right] \quad (21)$$

where $\eta_k \sim U(0, 1)$ are independently distributed uniform random variables. Now, note that the claim holds for $\varepsilon^* = 0$ since, in that case, we have that

$$P\left[\bigcap_k \{-\log(-\log(\eta_k)) \geq -\log(-\log(\eta_k) - \log(\eta_k)/[f_D(s_1, m')]_{t_k})\}\right] = 1,$$

using that $x \mapsto -\log(x)$ is strictly decreasing. Since all terms in Eq. 21 are continuous in $\varepsilon^*$, there exists $\varepsilon^*(m) > 0$, possibly dependent on $m$ but independent of $m'$, such that

$$P[R_m(\boldsymbol{U}, s_q) = R_{m'}(\boldsymbol{U}, s_q)] > P[R_m(\boldsymbol{U}, s_q) = R_{m'}(\boldsymbol{U}', s_q)] \quad (22)$$

holds if

$$\sup_{s_q} \|\log(f_D(s_q, m)) - \log(f_D(s_q, m'))\|_\infty < \varepsilon^*(m).$$

Since by assumption $D_t > 0$ for all $t \in V$, there exists $\varepsilon(m) > 0$ in probability space such that Eq. 22 holds if

$$\sup_{s_q} \|f_D(s_q, m) - f_D(s_q, m')\|_\infty < \varepsilon(m).$$

This concludes the proof.

### C.6 CALCULATION OF AVERAGE WIN-RATES IN THE EXAMPLE USED IN SECTION 1 AND APPENDIX B

In this section, we provide detailed calculations of the win-rates for the example in Section 1 and Appendix B. Recall that in this example, we are given three LLMs $m_1$, $m_2$ and $m_3$, and we need to rank them according to their ability to answer correctly two types of input prompts, $q$ and $q'$, picked uniformly at random. We assume that the true probability that each LLM answers correctly each type of input prompt is given by:

|  | $m_1$ | $m_2$ | $m_3$ |
|---|---|---|---|
| $q$ | $p_1 = 0.4$ | $p_2 = 0.48$ | $p_3 = 0.5$ |
| $q'$ | $p_1' = 1$ | $p_2' = 0.9$ | $p_3' = 0.89$ |

Using Proposition B.1, the win-rates under independent autoregressive generation are given, for each LLM $m_k$, by:

$$\frac{1}{2}\sum_{j \neq k} \mathbb{E}_{\boldsymbol{U}, \boldsymbol{U}' \sim P_U, S_q \sim P_Q}[\mathbf{1}\{R_{m_k}(\boldsymbol{U}, S_q) > R_{m_j}(\boldsymbol{U}', S_q)\}] = \frac{\sum_{j \neq k} p_k(1 - p_j) + \sum_{j \neq k} p_k'(1 - p_j')}{4}.$$

$$(23)$$

Substituting the numerical values we obtain:

$$
\frac{1}{2}\sum_{j\neq 1}\mathbb{E}_{\boldsymbol{U},\boldsymbol{U}'\sim P_{\boldsymbol{U}},S_q\sim P_Q}[\mathbf{1}\{R_{m_1}(\boldsymbol{U},S_q) > R_{m_j}(\boldsymbol{U}',S_q)\}] = 0.1545,
$$

$$
\frac{1}{2}\sum_{j\neq 2}\mathbb{E}_{\boldsymbol{U},\boldsymbol{U}'\sim P_{\boldsymbol{U}},S_q\sim P_Q}[\mathbf{1}\{R_{m_2}(\boldsymbol{U},S_q) > R_{m_j}(\boldsymbol{U}',S_q)\}] = 0.15675, \qquad (24)
$$

$$
\frac{1}{2}\sum_{j\neq 3}\mathbb{E}_{\boldsymbol{U},\boldsymbol{U}'\sim P_{\boldsymbol{U}},S_q\sim P_Q}[\mathbf{1}\{R_{m_3}(\boldsymbol{U},S_q) > R_{m_j}(\boldsymbol{U}',S_q)\}] = 0.16225
$$

Similarly, using Proposition B.1, the win-rates using coupled autoregressive generation can be written, for each LLM $m_k$, as:

$$
\frac{1}{2}\sum_{j\neq k}\mathbb{E}_{\boldsymbol{U}\sim P_{\boldsymbol{U}},S_q\sim P_Q}[\mathbf{1}\{R_{m_k}(\boldsymbol{U},S_q) > R_{m_j}(\boldsymbol{U},S_q)\}] = \frac{\sum_{j\neq k}(p_k - p_j)_+ + \sum_{j\neq k}(p'_k - p'_j)_+}{4},
$$

$$(25)$$

where $(\bullet)_+ = \max(0,\bullet)$ denotes the positive part. Substituting the numerical values we obtain:

$$
\frac{1}{2}\sum_{j\neq 1}\mathbb{E}_{\boldsymbol{U}\sim P_{\boldsymbol{U}},S_q\sim P_Q}[\mathbf{1}\{R_{m_1}(\boldsymbol{U},S_q) > R_{m_j}(\boldsymbol{U},S_q)\}] = 0.0525,
$$

$$
\frac{1}{2}\sum_{j\neq 2}\mathbb{E}_{\boldsymbol{U}\sim P_{\boldsymbol{U}},S_q\sim P_Q}[\mathbf{1}\{R_{m_2}(\boldsymbol{U},S_q) > R_{m_j}(\boldsymbol{U},S_q)\}] = 0.0225,
$$

$$
\frac{1}{2}\sum_{j\neq 3}\mathbb{E}_{\boldsymbol{U}\sim P_{\boldsymbol{U}},S_q\sim P_Q}[\mathbf{1}\{R_{m_3}(\boldsymbol{U},S_q) > R_{m_j}(\boldsymbol{U},S_q)\}] = 0.03.
$$

## D  ADDITIONAL EXPERIMENTAL DETAILS

**Hardware setup.** Our experiments are executed on a compute server equipped with $2 \times$ Intel Xeon Gold 5317 CPU, 1,024 GB main memory, and $2 \times$ A100 Nvidia Tesla GPU (80 GB, Ampere Architecture). In each experiment, a single Nvidia A100 GPU is used.

**Datasets.** As a benchmark dataset, we use Measuring Massive Multitask Language Understanding dataset (MMLU) Hendrycks et al. (2021a) consisting of 14,042 questions covering 52 diverse knowledge areas with each question offering four possible choices indexed from A to D, and a ground-truth answer. For pairwise comparison tasks, we use the first 500 questions from the LMSYS-Chat-1M dataset Zheng et al. (2024).

**Models.** In our experiments, we use `Llama-3.1-8B-Instruct`, its quantized variants `Llama-3.1-8B-Instruct-{AWQ-INT4, bnb-4bit, bnb-8bit}` and `Llama-3.2-{1B, 3B}-Instruct` models. The models are obtained from `Hugging Face`, and the quantised LLM variants `Llama-3.1-8B-Instruct-{bnb-4bit, bnb-8bit}` are built using the `bitsandbytes` library Bits and Bytes Foundation (2024).

**Prompts.** To instruct LLMs for generating output, we use the system prompt in Table 2 for the MMLU dataset and Table 3 for the LMSYS-Chat-1M dataset. Further, to perform pairwise comparisons of outputs of different LLMs, we use the system prompt in Table 4, which is adapted from Chiang et al. (2025), to prompt the strong LLM.

**System:** You will be given multiple choice questions. Please reply with a single character 'A', 'B', 'C', or 'D' only. DO NOT explain your reply.

Table 2: System prompt used for the MMLU dataset.

**System:** Keep your responses short and to the point.

Table 3: System prompt used for the LMSYS Chatbot Arena dataset.

**System:** Please act as an impartial judge and evaluate the quality of the responses provided by two AI assistants to the user prompt displayed below. Your job is to evaluate which assistant's answer is better. When evaluating the assistants' answers, compare both assistants' answers. You must identify and correct any mistakes or inaccurate information. Then consider if the assistant's answers are helpful, relevant, and concise. Helpful means the answer correctly responds to the prompt or follows the instructions. Note when user prompt has any ambiguity or more than one interpretation, it is more helpful and appropriate to ask for clarifications or more information from the user than providing an answer based on assumptions. Relevant means all parts of the response closely connect or are appropriate to what is being asked. Concise means the response is clear and not verbose or excessive. Then consider the creativity and novelty of the assistant's answers when needed. Finally, identify any missing important information in the assistants' answers that would be beneficial to include when responding to the user prompt. do not provide any justification or explanation for your response. You must output only one of the following choices as your final verdict:

'A' if the response of assistant A is better

'B' if the response of assistant B is better

'Tie' if the responses are tied

Table 4: System prompt used for obtaining pairwise preferences using `GPT-4o-2024-11-20` as the judge.

# E  ADDITIONAL EXPERIMENTAL RESULTS ON THE MMLU DATASET

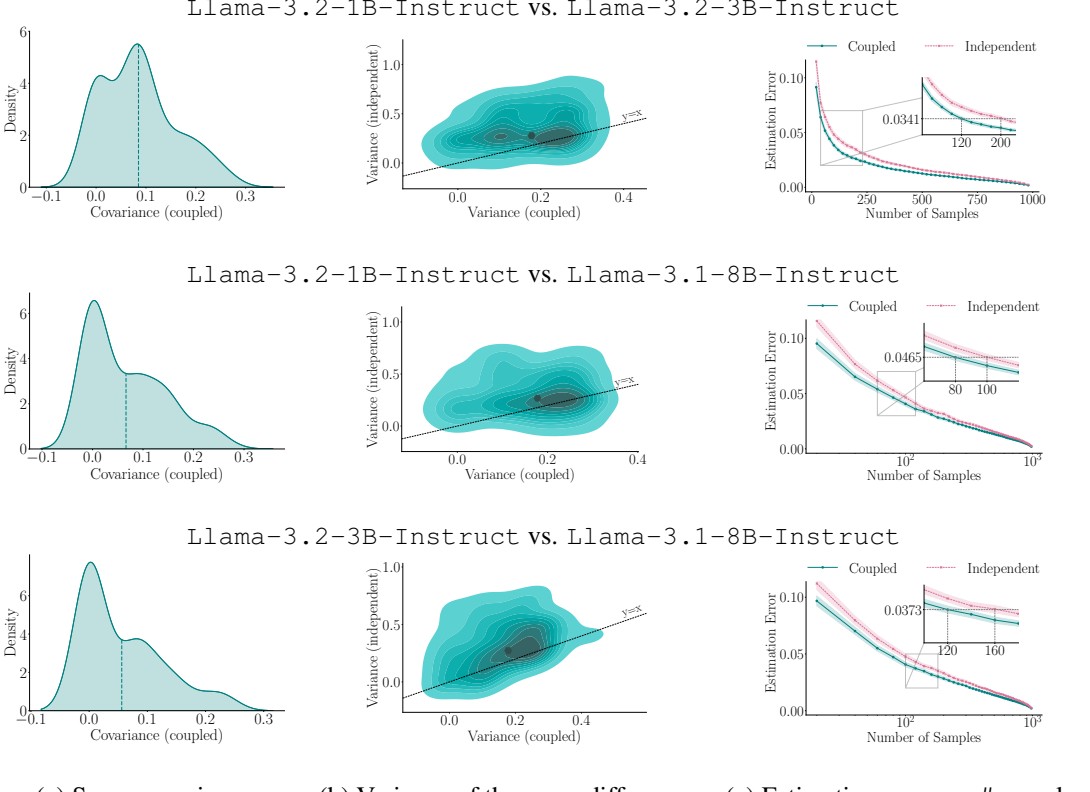

(a) Score covariance     (b) Variance of the score difference     (c) Estimation error vs. # samples

Figure 4: **Comparison between three pairs of LLMs on multiple-choice questions from the "college computer science" knowledge area of the MMLU dataset.** Panels in column (a) show the kernel density estimate (KDE) of the covariance between the scores of the two LLMs on each question under coupled generation; the dashed lines correspond to average values. Panels in column (b) show the KDE of the variance of the difference between the scores of the LLMs on each question under coupled and independent generation; the highlighted points correspond to median values. Panels in column (c) show the absolute error in the estimation of the expected difference between the scores of the LLMs against the number of samples; for each point on the x-axis, we perform $1,000$ sub-samplings and shaded areas correspond to $95\%$ confidence intervals.

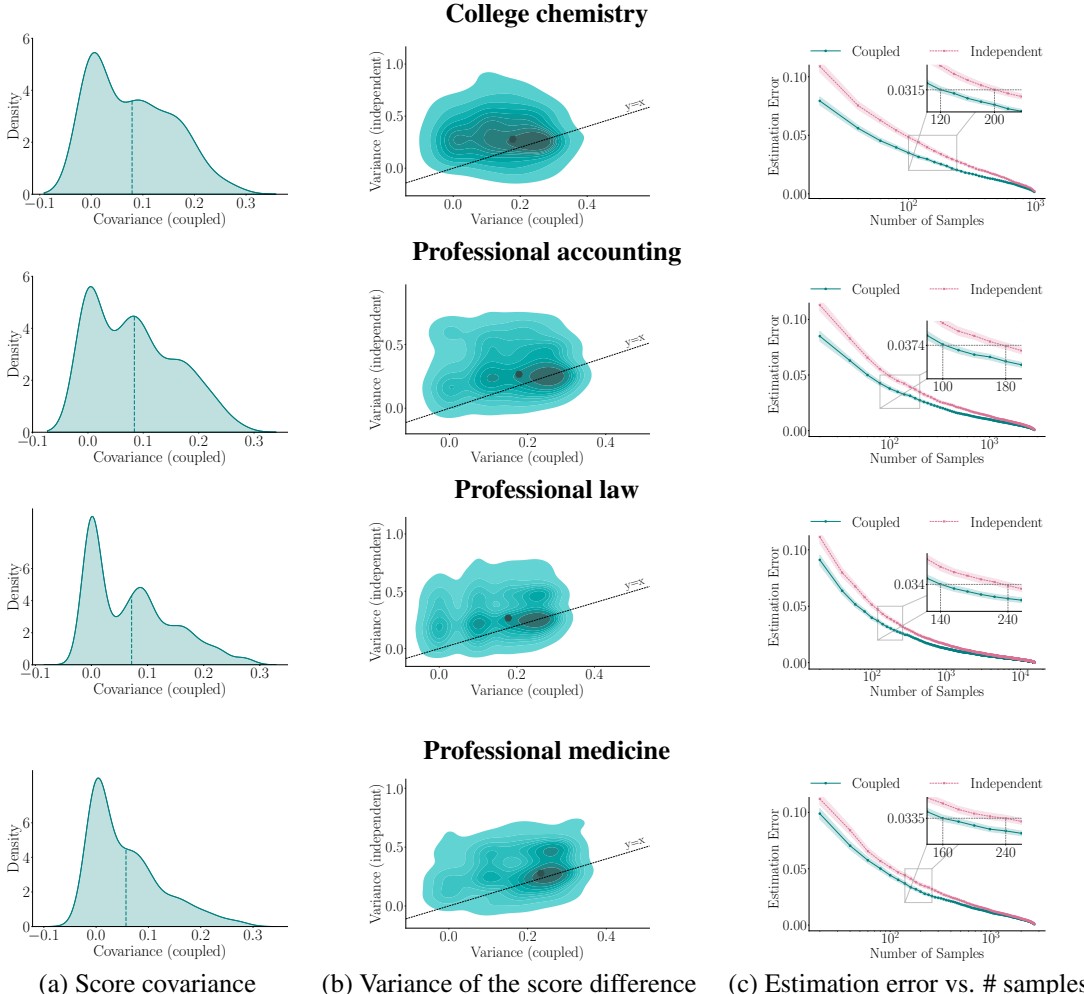

(a) Score covariance     (b) Variance of the score difference     (c) Estimation error vs. # samples

Figure 5: **Comparison between `Llama-3.2-1B-Instruct` and `Llama-3.2-3B-Instruct` on multiple-choice questions from four knowledge areas of the MMLU dataset.** Panels in column (a) show the kernel density estimate (KDE) of the covariance between the scores of the two LLMs on each question under coupled generation; the dashed lines correspond to average values. Panels in column (b) show the KDE of the variance of the difference between the scores of the LLMs on each question under coupled and independent generation; the highlighted points correspond to median values. Panels in column (c) show the absolute error in the estimation of the expected difference between the scores of the LLMs against the number of samples; for each point on the x-axis, we perform 1,000 sub-samplings and shaded areas correspond to 95% confidence intervals. We observe qualitatively similar results for other knowledge areas.

# F  ADDITIONAL EXPERIMENTAL RESULTS ON THE LMSYS-CHAT-1M DATASET

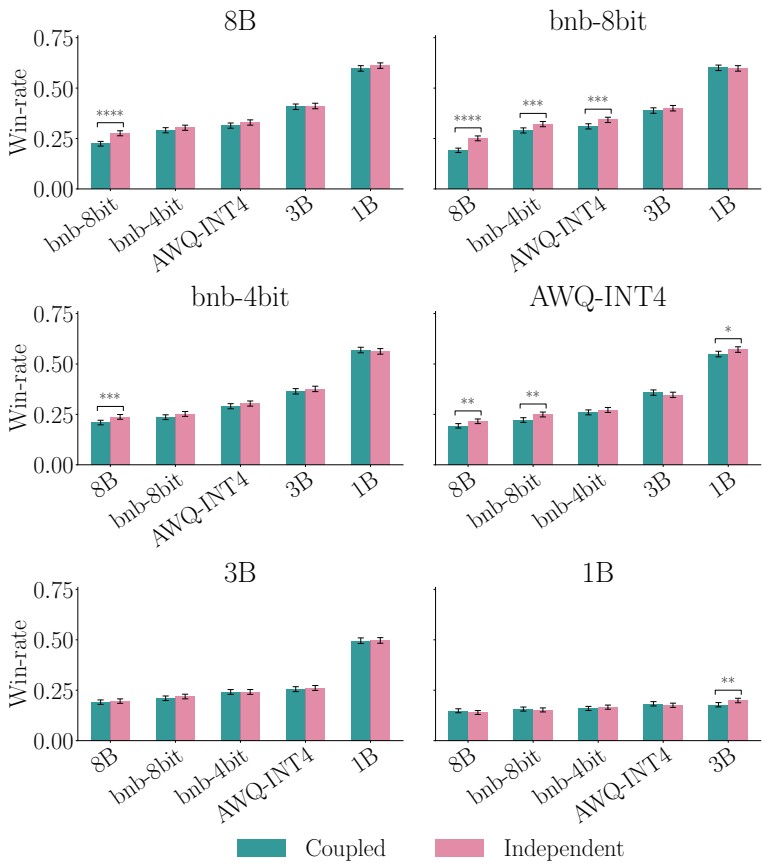

Figure 6: **Empirical win-rate of each LLM against other LLMs on questions from the LMSYS-Chat-1M dataset.** Empirical estimate of the win-rate under coupled autoregressive generation as given by Eq. 8 and under independent generation generation as given by Eq. 7. Each empirical win-rate is computed using pairwise comparisons between the outputs of each LLM and any other LLM over 500 questions with 10 (different) random seeds. The error bars correspond to 95% confidence intervals. For each pair of empirical win-rates, we conduct a two-tailed test, to test the hypothesis that the empirical win-rates are the same; (∗∗∗∗, ∗∗∗, ∗∗, ∗) indicate $p$-values ($< 0.0001$, $< 0.001$, $< 0.01$, $< 0.05$), respectively.

