# OpenReview forum: "Evaluation of Large Language Models via Coupled Token Generation"
_ICLR.cc/2025/Workshop/BuildingTrust — BuildingTrust_

### Official Review · Reviewer_S3Ha · 2025-02-17
**Review of Large Language Models via Coupled Token Generation**

**Rating:** 6
**Confidence:** 3

**Review:**

Summary
The paper introduces a causal model for coupled autoregressive generation, which shares the same randomness across different LLMs to ensure that performance differences are due to model characteristics rather than sampling noise. The authors theoretically demonstrate that this approach reduces the variance of performance estimates, leading to more sample-efficient evaluations and potentially different model rankings in pairwise comparisons. Empirical results on benchmarks MMLU and LMSYS Chatbot Arena further validate that coupled generation provides more reliable and intuitive assessments of LLM performance compared to independent sampling.

Strengths
Approach is both theoretically and empirically justified.
Highlights an important, underexplored source of variability—sampling randomness—shows how it can confound model evaluation, and offers a solution.

Weaknesses/Questions
The practical details of coupling the noise across models are not thoroughly detailed, leaving some ambiguity in replication. Details could be added to the appendix.

Minor comments:
Lines 121, 194 1099, 1101 contains broken ?? equation or section references.

---

### Official Review · Reviewer_7LdU · 2025-02-20
**good evalution method but a bit complex**

**Rating:** 6
**Confidence:** 3

**Review:**

The paper proposes an evaluation method that reduces the randomness in LLMs evaluation through coupled generation. This approach is theoretically innovative, especially in offering a new perspective on addressing the inconsistency in LLM evaluations caused by randomness. The paper demonstrates that coupled generation can reduce the need for samples while maintaining the reliability of evaluation results.

However, while the causal model provides a framework for understanding randomness, its application in the paper seems somewhat complex. The core idea of sharing random seeds could potentially be explained and implemented in a simpler manner.

The theoretical analysis relies on "canonical settings" such as binary choice questions, single-token responses, and the Gumbel-Max SCM. These settings might be overly simplified and may not adequately represent real-world LLM application scenarios, limiting the generalizability of the results.

Additionally, using GPT-4 as the judge for evaluation seems weak. At a minimum, GPT-4 Turbo would offer better performance for this task.

Lastly, the causal model's explanation appears unclear at times, potentially hindering readers from fully grasping the paper's proposed method. A clearer and more accessible explanation would benefit the overall presentation.

---

### Decision · Program_Chairs · 2025-03-05

Accept